

# Phylogenetic relationships and genetic diversity of the *Polypedates leucomystax* complex in Thailand

Kittisak Buddhachat[1,*] and Chatmongkon Suwannapoom[2,*]

[1] Department of Biology, Naresuan University, Phitsanulok, Thailand
[2] School of Agriculture and Natural Resources, University of Phayao, Phayao, Thailand
[*] These authors contributed equally to this work.

## ABSTRACT

Taxonomic uncertainty of the Asian tree frog *Polypedates leucomystax* complex presents the challenging task of inferring its biogeographical history. Here, we describe its dispersion and the genetic relationships among different populations in Thailand, where we connect the population of the *P. leucomystax* complex of the Sunda Islands to the Indochina (mainland) population based on analyses of 266 sequences of the mitochondrial cytochrome c oxidase subunit I (*COI*) gene. Our maternal genealogy implies that there are four well-supported lineages in Thailand, consisting of Northern A (clade A: *Polypedates* sp.), Nan (clade B: *P.* cf. *impresus*), Southern (clade C: *P.* cf. *leucomystax*) and Northern D (clade D: *P.* cf. *megacephalus*), with Bayesian posterior probability >0.9. Phylogeny and haplotype networks indicate that clades A, B and D are sympatric. In contrast, clade C (*P.* cf. *leucomystax*) and clade D (*P.* cf. *megacephalus*) are genetically divergent due to the geographical barrier of the Isthmus of Kra, resulting in an allopatric distribution. Climatic conditions, in particular differences in rainfall on each side of the Isthmus of Kra, may play an important role in limiting the immigration of both clades. For the within-populations of either clades C or D, there was no significant correlation between geographic and genetic distance by the isolation-by-distance test, indicating intraspecific-dispersal of each clade. Population expansion occurred in clade C, whereas clade D showed a constant population. Taken together, the *P. leucomystax* complex in South East Asia may have diversified under climatic pressure, leading to allopatric and/or sympatric speciation.

## INTRODUCTION

Southeast Asia contains a substantial genetic diversity of amphibians. Recent molecular phylogenetic analyses have disclosed many anuran lineages that contain cryptic species. Historically, complex changes in the region's geology and climate (e.g., Pleistocene climatic oscillations) altered the topology and environmental conditions, resulting in an initial fragmentation of habitat. These mechanisms generated the high species richness observed today (*Hall, 1998*; *Woodruff, 2010*). Of interest to our research were the numerous frog species in Southeast Asia whose taxonomy is still controversial, such as *Microhyla fissipes*

Corresponding author
Kittisak Buddhachat,
kittisakbu@nu.ac.th,
k_buddhachat@yahoo.com

(*Yuan et al., 2016*), *Staurois tuberilinguis* (*Matsui et al., 2007*), *Microhyla ornata* (*Matsui et al., 2005*) and *Polypedates leucomystax* (*Kuraishi et al., 2013*; *Rujirawan, Stuart & Aowphol, 2013*). Accurate species delimitation is essential to better understand their speciation and diversification and their biogeography for conservation purposes.

The Asian tree frog, the *P. leucomystax* (Gravenhorst, 1829) complex, is an Asian Rhacophoridae frog. These species are widely distributed in Southeast Asia, South China and India. In addition, this species has phenotypic plasticity and exhibits high adaptation to the local environment, leading to its existence in diverse habitats such as forests and even buildings. These high levels of phenotypic plasticity present a great challenge for classification. Phylogenetic and taxonomic relationships of the *P. leucomystax* complex throughout Southeast Asia exhibit adaptive radiation (*Kuraishi et al., 2013*; *Pan et al., 2013*; *Rujirawan, Stuart & Aowphol, 2013*). At least six valid species, including *P. braueri*, *P. impresus*, *P. leucomystax*, *P. macrotis*, *P. megacephalus* and *P. mutus* have been distinguished in the *P. leucomystax* complex based on their morphology, advertisement calls and molecular data (*Matsui, Seto & Utsunomiya, 1986*; *Brown et al., 2010*; *Kuraishi et al., 2011*; *Kuraishi et al., 2013*; *Pan et al., 2013*). Five species of genus *Polypedates*, *P. colletti*, *P. leucomystax*, *P. macrotis*, *P. megacephalus* and *P. mutus*, can be found in Thailand (*Taylor, 1962*; *Heyer, 1971*; *Frost, 2013*; *Kuraishi et al., 2013*; *Pan et al., 2013*; *Rujirawan, Stuart & Aowphol, 2013*). *Brown et al. (2010)* indicated that much of the genetic divergence of the *P. leucomystax* complex was discovered in mainland rather than insular populations distributed throughout thousands of islands of the Malay Archipelago, presumably resulting from range expansion mediated by transportation of agricultural products. Recently, a new species, *P. discantus*, belonging to the *P. leucomystax* species complex from southern Thailand was discovered using data on morphological characteristics, advertisement calls and molecular evidence, which showed that this species was highly dissimilar to *P. leucomystax* and *P. megacephalus* (*Rujirawan, Stuart & Aowphol, 2013*). Several studies have confirmed the existence of cryptic species of the *P. leucomystax* complex (*Matsui, Seto & Utsunomiya, 1986*; *Kuraishi et al., 2011*; *Blair et al., 2013*; *Kuraishi et al., 2013*; *Pan et al., 2013*).

In this study, we investigated the genetic variation, phylogenetic relationships and other relevant factors that limit the dispersal of the *P. leucomystax* complex in Thailand. The present results illustrate the range of distribution of putative *P. leucomystax* and putative *P. megacephalus*, which is influenced by climatic conditions.

## MATERIALS AND METHODS
### Sample collection, DNA extraction and sequencing
A total of 266 adult *P. leucomystax* complex individuals were collected from 15 different localities in Thailand (Table 1). All samples were dissected to obtain the liver, which was then stored in absolute ethanol. Sample collection and euthanization were approved by the Center For Animal Research Naresuan University under project number NU-AE591028. Genomic DNA was extracted from liver tissue using a DNA extraction kit (RBC Bioscience, Singapore) and kept at −20 °C for further use. Individual DNA was used as a template for

**Table 1** Localities of sample collection for *Polypedates leucomystax* complex in Thailand.

| Locality | Abbreviation | Number | Altitude (m above sea level) | Longitude | Latitude |
|---|---|---|---|---|---|
| Nan province | NAN | 12 | 665 | 18.980974 | 101.182594 |
| Kanchanaburi province | KCB | 20 | 917 | 14.69329 | 98.40535 |
| Loei province: Phu Ruea | LPR | 11 | 939 | 17.48193 | 101.34982 |
| Nakhon Ratchasima province | NRS | 14 | 865 | 14.49336 | 101.87364 |
| Chiang Mai province: Mae Wang | CM | 7 | 678 | 18.657305 | 98.681831 |
| Chiang Mai province: Doi Saket | CM | 13 | 402 | 18.98777 | 99.11455 |
| Chiang Mai province: Omkoi | CM | 13 | 460 | 17.47137 | 98.45785 |
| Mae Hong Son province | MHS | 44 | 396 | 19.24797 | 97.99542 |
| Saraburi province | SRB | 12 | 105 | 14.70993 | 100.81819 |
| Phetchaburi province | PCB | 22 | 329 | 14.70993 | 100.81819 |
| Prachuap Khiri Khan province | PKK | 15 | 23 | 11.43678 | 99.56011 |
| Ranong province | RN | 14 | 18 | 9.6052 | 98.4669 |
| Nakhon Si Thammarat province | NST | 37 | 97 | 8.76902 | 99.80349 |
| Phuket province: Thalang | PK | 17 | 31 | 7.96804 | 98.33589 |
| Chumphon province | CP | 15 | 103 | 10.110278 | 99.082778 |

PCR amplification of the mitochondrial COI gene using Taq DNA polymerase in a total volume of 25 μL under the following conditions: an initial denaturation at 94 °C for 5 min, followed by 35–40 cycles at 94 °C for 30 s, 50 °C for 30 s and 72 °C for 1 min, and a final extension step at 72 °C for 7 min. PCR products were visualized on 1.5% agarose gel under UV illuminator. The expected size of the partial mitochondrial *COI* gene sequence was 688 bp. Subsequently, all PCR products were purified using a QIAquick PCR Purification Kit (Qiagen, Hilden, Germany) and then sequenced (Macrogen, Seoul, South Korea).

## Phylogeny

Bayesian inference (BI) and maximum likelihood (ML) were employed for constructing a phylogenetic tree based on the following partial *COI* sequences of the *P. leucomystax* complex that were retrieved from GenBank: *P. impresus*: KP996822 (China), KP996846 (China), KP087862–KP087870 (Laos); *P. leucomystax*: KR087871–KR087872 (Thailand); and *P. megacephalus*: KR087879, KR087881 (Thailand). First, the best-fit model of DNA sequence evolution for this locus was identified with the Akaike information criterion (AIC) implemented in MrModeltest v2.3 (*Nylander, 2004*), resulting in the GTR+I+G model as the best fit with AIC. Subsequently, a Bayesian tree was constructed based on the base substitution calculated from the GTR+I+G model through MrBayes 3.1.2 (*Ronquist & Huelsenbeck, 2003*) with two independent searches with random starting trees for five million generations, in which the diagnostic was calculated every 1,000 generations and compared using four Markov chain Monte Carlo chains (temp = 0.2). The log-likelihood scores were used for plotting the convergence in Tracer v1.5 (*Rambaut et al., 2013*) and building a consensus tree, which was completed by the removal of the first 25% of the generations from each run. For maximum likelihood analysis, RAxML 7.0.4 was carried

out (*Stamatakis, Hoover & Rougemont, 2008*) using the GTR+I+G model for nucleotide substitution (same as the BI analysis) with 1,000 bootstrap replicates.

## Population genetics and structure

A total of 266 sequences of mitochondrial *COI* were aligned using ClustalW (implemented in MEGA 6.0 with default parameters). The number of polymorphic sites, the parsimony-informative sites, singleton sites, the number of haplotypes, haplotype diversity ($H_d$), and nucleotide diversity for each clade were calculated using DnaSP v5.0 (*Librado & Rozas, 2009*). Genetic distances among taxa were calculated using the corrected *p*-distance model in MEGA 6.0 (*Tamura et al., 2013*). Furthermore, we detected a boundary line in the genetic landscape between the Northern D clade and the Southern clade using Barrier 2.2 (*Manni, Guérard & Heyer, 2004*). A minimum spanning network was constructed using PopART (Population Analysis with Reticulate Trees) to define the relationships among haplotypes and the distribution of haplotypes in each locality (*Bandelt, Forster & Röhl, 1999*). To evaluate the effect of geographic distance on the genetic divergence among populations of the Northern D clade and among populations of the Southern clade, a linear regression model was carried out.

## Demographic history

To investigate the demographic history of *P. megacephalus* and *P. leucomystax* populations in Thailand, multiple approaches were explored using DnaSP (*Librado & Rozas, 2009*). Neutrality tests of Tajima's *D* (*Tajima, 1989*) and Fu's *Fs* (*Fu, 1997*) for the two species were completed. A significantly positive value indicates a process of subdivision or a recent population bottleneck, whereas a population expansion results in a significantly negative value. Pairwise mismatch distribution was used assuming a constant population size (*Rogers & Harpending, 1992*). Multimodal mismatch distribution implies stability of the population, while unimodal mismatch distribution reflects an expanding population. In addition to these methods, the raggedness index ($r_g$) of the observed distribution was calculated (*Harpending, 1994*). A small $r_g$ indicates a demographic expansion.

# RESULTS

## Sequence characteristics

A total of 266 samples of the *P. leucomystax* complex yielded 688 bp fragments of the mitochondrial *COI* gene. All new sequences in this study were deposited in the GenBank database (MG583020–MG583285). After multiple alignment of all *COI* sequences, the sequences were trimmed to the same length (437 bp) before downstream analysis. We observed 82 polymorphic sites, which are also 82 parsimony-informative sites without a singleton site, resulting in the acquisition of 15 haplotypes (Table 2). Overall nucleotide and haplotype diversity were 0.0664 and 0.9000, respectively (Table 2).

## Phylogenetic analyses and haplotype distribution

Based on 266 mitochondrial *COI* sequences of the *P. leucomystax* complex, a matrilineal genealogy was generated, and our results indicated that the *P. leucomystax* complex in

**Table 2 Summary of the *P. leucomystax* complex in Thailand.** Major lineages clades, putative scientific name, number of individuals ($N$), number of mtDNA haplotypes ($n$), number of polymorphic sites (P), parsimony-informative sites (PI) and singleton sites (S), haplotype diversity ($H_d$) and nucleotide diversity ($\pi$).

| Clade | Putative species | $N$ | $n$ | $\pi$ | $H_d$ | P | S | PI |
|---|---|---|---|---|---|---|---|---|
| A (the Northern A) | *Polypedates* sp. | 40 | 2 | 0.0037 | 0.4089 | 4 | 0 | 4 |
| B (Nan) | *P . impresus* | 12 | 1 | 0 | 0 | 0 | 0 | 0 |
| C (the Southern) | *P . megacephalus* | 131 | 7 | 0.0048 | 0.746 | 15 | 1 | 14 |
| D (the Northern D) | *P . leucomystax* | 83 | 5 | 0.0073 | 0.7526 | 7 | 0 | 7 |
| Total | | 266 | 15 | 0.0664 | 0.9 | 82 | 0 | 82 |

Thailand consists of four clades: clade A (Northern A), *Polypedates* sp.; clade B (Nan), *P.* cf. *impresus*; clade C (Southern), *P.* cf. *leucomystax*; and clade D (Northern D), *P.* cf. *megacephalus* (Fig. 1). With respect to phylogenetic inference, clade A was treated as a sister group of clade B, which was found in Nan. Clade A, however, can be seen in genetic samples obtained from the Kanchanaburi (KCB), Mae Hong Son (MHS) and Phetchaburi (PCB) provinces and shared a habitat with clade D, which was recognized as *P.* cf. *megacephalus*; its distribution range was in the far north of the Isthmus of Kra at Chiang Mai (CM), MHS, KCB, PCB, Saraburi (SRB), Loei (LPR), Nakhon Ratchasima (NRS) and Prachuap Khiri Khan (PKK). The dispersal areas of clade C, as represented by *P.* cf. *leucomystax*, included Chumphon (CP), Nakhon Si Thammarat (NST), Phuket (PK) and Ranong (RN), which are south of the Isthmus of Kra (Fig. 2A). Barrier 2.2 was employed to determine a barrier for immigration between clade C and clade D populations based on the dataset of genetic distance (Kimura's two-parameter model). Likely, the Isthmus of Kra (IOK) represents a significant barrier to restrict immigration based on the great genetic distance that was noted around IOK (Fig. 2B).

A minimum spanning network among the mitochondrial haplotypes was also constructed as shown in Fig. 1. Clade D exhibited the highest number of haplotypes at seven ($H_d = 0.746$, followed by clade C with five haplotypes ($H_d = 0.7526$) (Table 2). Haplotypes A and B, seen in clade A, and haplotype C found in clade B were unique haplotypes. Haplotypes D–I were noted in clade D, while populations of northern, western and upper southern Thailand (KCB, PCB and PKK, respectively) shared haplotype J. Haplotypes F and G of clade D are considerably divergent from the rest of clade D and we partition it in two subclades: D1 and D2. Clade C (*P.* cf. *leucomystax*) had high haplotype diversity and contained three unique haplotypes (M, N and O) and two shared haplotypes (K and L).

Analyses of the linear regression model between the genetic distance of the mitochondrial *COI* gene sequence and the geographical distance found no significant isolation-by-distance effect among populations of clade C (*P.* cf. *leucomystax*) and clade D (*P.* cf. *megacephalus*) (Fig. 3).

## Demographic history

When we defined a significant barrier around the Isthmus of Kra (IOK) leading to the genetic divergence between clade C (*P.* cf . *leucomystax*) and clade D (*P.* cf. *magacephalus*),

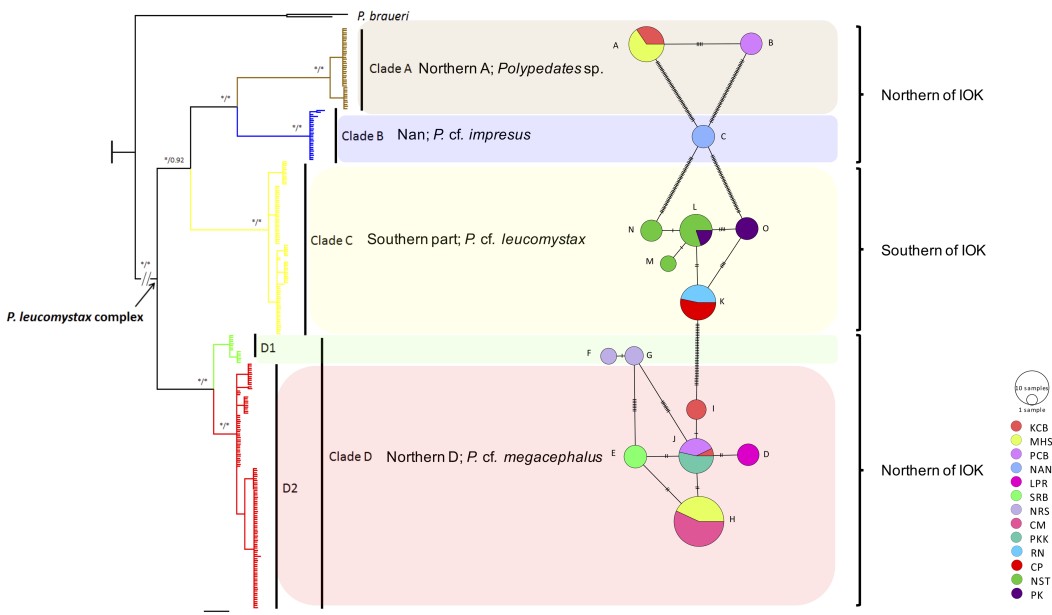

**Figure 1** **Phylogeographic relationships and a minimum spinning network of all haplotype of the** *Polypedates leucomystax* **complex among populations in Thailand.** It consists of clade A as *Polypedates* sp., clade B as *P.* cf. *impresus*, clade C (the Southern clade) as *P.* cf. *leucomystax*, and clade D (the Northern D clade) as *P.* cf. *megacephalus*, as well as outgroups (KR087858, KP996762 = *P. braueri*) inferred from Bayesian analysis of mitochondrial COI gene sequences. Bayesian posterior probability values are expressed above internodes. The asterisks above branches represent bootstrap support for Bayesian posterior probabilities and maximum likelihood (>95%). Scale bar represents 0.5 nucleotide substitutions per site. IOK represents the Isthmus of Kra.

neutrality tests (Tajima's *D* and Fu's *Fs*) of both species were not significantly positive, whereas Fu's *Fs* of clade C was significantly positive (Table 3). Furthermore, the mismatch distribution was tested as a result of a left-skewed multimodal mismatch distribution for clade D with moderate $r_g$ (0.2031) but a unimodal mismatch distribution for clade C with a low $r_g$ (0.0569) (Fig. 4). Overall, these results suggested a constant population size of clade D and a population expansion of clade C.

## DISCUSSION

The taxonomy of the Asian tree frog of the *P. leucomystax* complex is contentious due to the species' widespread distribution from Nepal to South East Asia and similar morphologies. To better understand the population structure and biogeography of the *P. leucomystax* complex in Thailand, where there is a substantial area for their genetic dispersal, the *COI* mitochondrial gene sequences of these species were analysed. Our matrilineal genealogy implied four well-supported lineages, consisting of a Northern D clade (clade D), a Southern clade (clade C), a Nan clade (clade B) and a Northern A clade (clade A); based on their sequences and distributions, they might be treated as *P. megacephalus*, *P. leucomystax*, *P. impresus* and *Polypedates* sp., respectively.

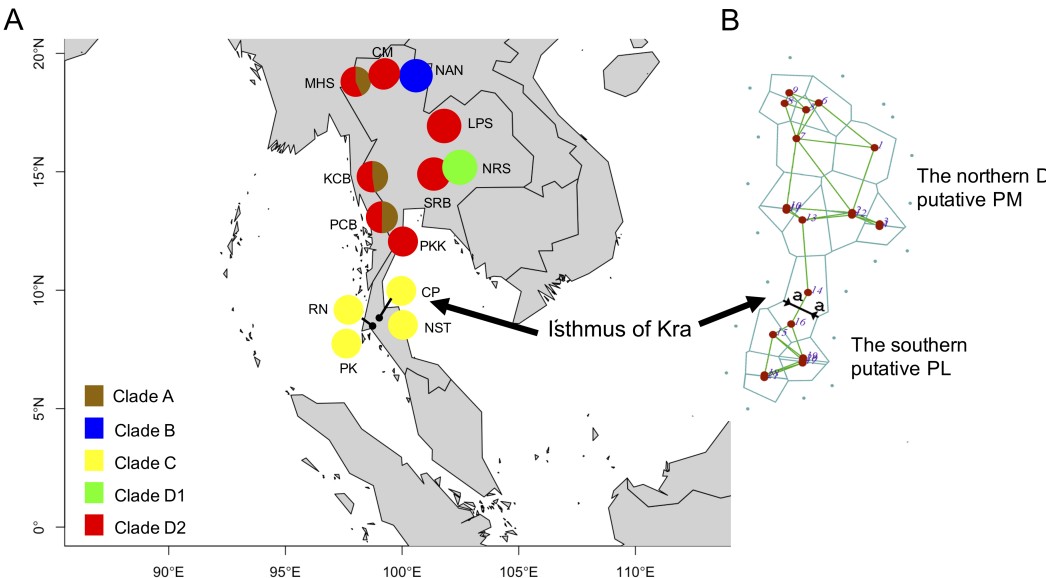

**Figure 2** Clade distribution of the *P. leucomystax.* complex throughout Thailand (A) and the genetic differentiation across the Northern D clade (putative *P. megacephalus*; PM) and the Southern clade (putative *P. leucomystax*; PL) (B). The abbreviations for each locality are given in Table 1. Different colors represent the different clades. "a" represents a significant barrier to partition the distribution of the two clades, by Barrier version 2.2.

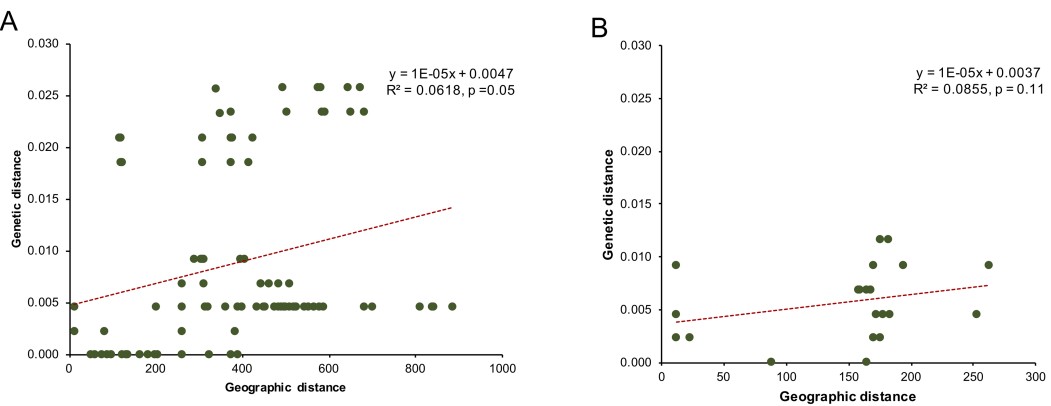

**Figure 3** The correlation of genetic distance and linear geographic distance (km) for (A) the Northern D clade (putative *Polypedates megacephalus*) and (B) the Southern clade (putative *Polypedates leucomystax*).

Although the mtDNA phylogeny reveals the presence of distinct clades, mtDNA introgression is frequently observed where closely related species are in secondary contact (*Toews & Brelsford, 2012*; *Zieliński et al., 2013*; *Wielstra et al., 2017a*; *Wielstra et al., 2017b*). This can occur particularly in sympatric populations or at the boundaries of species distributions resulting from incomplete reproductive isolation. For instance, a pair of crested newt species (genus *Triturus*) form a hybrid zone (south of the Marmara

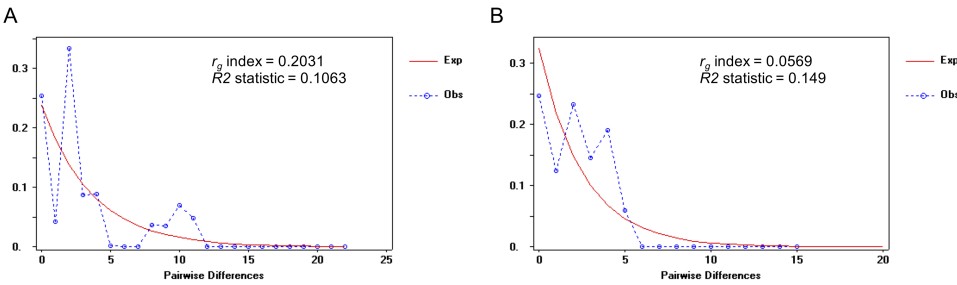

**Figure 4 Mismatch distribution of the mitochondrial COI gene in (A) the Northern D clade (putative *Polypedates megacephalus*) and (B) the Southern clade (putative *Polypedates leucomystax*).** The raggedness (rg) index is calculated to evaluate the population expansion of each species. Ramos-Onsins and Rozas's R2 statistic represents the population growth.

**Table 3 Summary of statistics used to compute the demographic history of populations of the Northern B clade (putative *P. megacephalus*) and the Southern clade (putative *P. leucomystax*).**

| Clade | Tajima's $D$ | | Fu's $Fs$ | |
|---|---|---|---|---|
| | $D$ | $P$ value | $Fs$ | $P$ value |
| The Northern D | 0.439 | >0.1 | 3.213 | 0.045 |
| The Southern | 1.176 | >0.1 | 3.031 | 0.071 |

Sea), where introgression suggests movement of the hybrid zone (*Wielstra et al., 2017b*). Therefore, the use of mitochondria DNA data alone is not sufficient for species delineation. An integrative approach, consulting additional data such as morphology, bioacoustics, ecology, and/or nuclear DNA, would be required to more accurately assess the taxonomy of the *P. leucomystax* complex (*Padial et al., 2010*). *Kuraishi et al. (2013)* found one sample of the *P. leucomystax* complex from Indochina shared nuclear brain-derived neurotrophic factor (BDNF) and recombination activating gene 1 (RAG-1) haplotypes with one sample from the south of IOK, possibly suggesting deep coalescence or incomplete lineage sorting.

Bayesian inference strongly supported the geographic distinction of species in the Northern D clade and the Southern clade. Furthermore, Monmonier's algorithm suggested that the Isthmus of Kra (IOK; located between 11 and 13°N along the Thai Peninsula) corresponds to a considerable phylogeographic break as evidenced by the large genetic divergence between the populations on either side of the isthmus. *Kuraishi et al. (2013)* explored the population structure of *P. leucomystax* complex in Southeast Asia and revealed that *P. megacephalus* is restricted to Indochina (the northern Thailand, Laos, Vietnam and southern China) (*Kuraishi et al., 2013*; *Pan et al., 2013*), while the range of *P. leucomystax* is restricted to the Sunda regions (southern Thailand and Malay Archipelago) (*Brown et al., 2010*; *Kuraishi et al., 2013*). Based on their distribution, we ascribe the Northern D clade to *P. megacephalus* and the Southern clade to *P. leucomystax*. The Isthmus of Kra has been well-characterized as a considerable biogeographic boundary of both faunal and floral assemblages that are limited on either side. *De Bruyn et al. (2005)* used genetic evidence from *Macrobrachium rosenbergii* (freshwater shrimp with a broad distribution) to

hypothesize that the existence of a seaway across the IOK over a million year contributed to its geographically discontinuous distribution. In contrast, *Hughes et al. (2011)*, based on marine fossils and geological data, proposed that the occurrence of species with distinct distributions in the north and south of the IOK is related to the influence of the climatic zone rather than either marine inundation or breaches at any point near the IOK. In addition, flying animals such as birds and bats also have a restricted species distribution which is bounded by the IOK (*Hughes et al., 2011*). This indicates that the geophysical barriers at the IOK, such as the waterway, are unlikely to be significant biogeographical boundaries causing vicarrance of various species at the IOK. *Hughes et al. (2011)* also determined the climatic zone in Indochina and the Malay Peninsula is the result of four climatic zonations, which are displayed as similar climatic zones found in each time period from the Last Interglacial Period to the current conditions. Of interest is the climatic divergence in precipitation, temperature and seasonality between the central zone (north of IOK) and the southern zone (south of IOK). Based on this, we suggest climate was a significant factor in shaping the spatial distribution of the genetic lineage between the Northern and Southern clades. The haplotype network also shows that the Northern D clade and the Southern clade have the highest genetic diversity and a wide range distribution in Thailand, although the most recent divergence time between *P. megacephalus* (putative Northern D clade) and *P. leucomystax* (putative Southern clade) was estimated to be in the late Pliocene or early Pleistocene (1.4–4.0 MYBP) (*Kuraishi et al., 2013*). We assumed that the populations of the Southern clade (putative *P. leucomystax*) expanded and colonized the northern part of Indochina, leading to the diversification of the species. This hypothesis is partially supported by the results of the unimodal pairwise difference and the small value of the raggedness index in the Southern clade (Fig. 3).

Within the population of the Northern D clade, maternal genealogy demonstrated that the genetic samples from Nakhon Ratchasima province (NRS) seemed to represent a naturally occurring divergence because of the emergence of endemic haplotypes; however, it was a low-supported lineage, with 0.7 Bayesian posterior probability (BPP). When we considered the topography of this region, the population of NRS as clade D1 is partitioned from the other populations within clade D by the two mountain ranges, Dong Phaya Yen and Sankamphaeng. This may be a possible barrier to gene flow among the modern populations of Northern D between eastern (clade D1) and western (clade D2; NRS) Thailand. Unfortunately, only a limited number of populations from the east of Thailand were investigated in this study. Further work for validating whether Dong Phaya Yen and Sankamphaeng are a barrier to gene flow in *P. megacephalus* is required. According to the demographic history, the population of Northern D was a stable population, but the population of Southern clade showed a relatively similar unimodal distribution with a small raggedness index, possibly indicating a population expansion. This result was similar to that observed in the population of *P. leucomystax* in the northern Philippines (*Brown et al., 2010*). This scenario implied a genetically homogenous population, especially in the population of Phuket Island, which shared a haplotype with NST, probably caused by a recent population expansion due to the founder effect. Although the population of the Northern D clade expanded, it was limited to localities south of the Isthmus of Kra.
## CONCLUSIONS

Our matrilineal genealogy of the *P. leucomystax* complex in Thailand suggested four lineages, i.e., Nan (putative *P. impresus*), Northern D (putative *P. megacephalus*), Southern (putative *P. leucomystax*) and Northern A (*Polypedates* sp.) clades. We noted that the populations of the Northern D, Nan and *Polypedates* sp. clades are in sympatry, while their distributions are allopatric to the southern clade (*P. leucomystax*) due to the separation by the Isthmus of Kra. Climatic conditions may be a major contributor to limited migration of the current populations of both clades, but climatic oscillation in the Pliocene and Pleistocene is a highly possible scenario that drove speciation resulting in diversification of the *P. leucomystax* complex in Southeast Asia and China, which includes the divergence of the southern and northern clades in Thailand.

### Funding
This work was supported by research funding from Naresuan University, Phitsanulok, Thailand (No. R2560C166) and the Thailand Research Fund (TRF) (DBG6180001). The funders had no role in study design, data collection and analysis, decision to publish, or preparation of the manuscript.

### Grant Disclosures
The following grant information was disclosed by the authors:
Naresuan University, Phitsanulok, Thailand: R2560C166.
Thailand Research Fund: DBG6180001.

### Competing Interests
The authors declare there are no competing interests.

### Author Contributions
- Kittisak Buddhachat and Chatmongkon Suwannapoom conceived and designed the experiments, performed the experiments, analyzed the data, contributed reagents/materials/analysis tools, wrote the paper, prepared figures and/or tables, reviewed drafts of the paper.

### Animal Ethics
The following information was supplied relating to ethical approvals (i.e., approving body and any reference numbers):
Sample collection and euthanasia were approved by the Center For Animal Research Naresuan University under project number NU-AE591028.

### DNA Deposition
The following information was supplied regarding the deposition of DNA sequences:
The sequence were supplied in a Supplemental file and also deposited in GenBank under accession numbers MG583020 to MG583285.

## Supplemental Information

Supplemental information for this article can be found online at http://dx.doi.org/10.7717/peerj.4263#supplemental-information.

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
