# Peer review of "Phylogenetic relationships and genetic diversity of the Polypedates leucomystax complex in Thailand"

_PeerJ, doi:10.7717/peerj.4263_

## Round 0.1 · original submission · Major Revisions

The main issue with the current ms is that the authors tend to equate mtDNA clades with species. While this may be likely, there are now many examples of deep mtDNA splits within clean biological species, either through hybridisation/introgression, maybe many generations back, leading to "ghost lineages", or simply retention of ancient lineages due to high Ne or population structure (splitting then merging). The authors should be focusing on the structure shown and what that means phylogeographically, POSSIBLY suggesting new species, but requiring further morphometric, behavioural or nuclear gene data.

Both reviewers comment on the writing/language, which needs some attention based on their helpful reviews.

The direction, aims and hypotheses need to be made clearer at the outset, as well as the number of described species and subspecies in Thailand. The importance of the Isthmus of Kra could be emphasised.

With respect to the analysis, you should use corrected distance p values following the model that you used for phylogenetic analysis. The Mantel test has better alternatives.

Reviewer 1 ·

Basic reporting

No comment

Experimental design

No comment

Validity of the findings

No comment

Additional comments

Buddhachat and Suwannapoom conduct a phylogeography of frogs belonging to the Polypedatus leucomystax complex occurring in Thailand. There is a basis for an interesting paper here, but in current form I find it unfocussed. I find it difficult from the present paper to get a good idea about this complex, not only in general, but also the situation in Thailand. Particularly relevant for this study is that it is not clear how many species are supposed to occur in Thailand and if they are all included. The aim of the paper is not clear. Obviously the presence of cryptic species in the complex and potential co-occurrence of such species is a big part of it. However, the paper suffers from equating mtDNA clades with species. I suggest to tone this way down and merely suggest that mtDNA clades might suggest additional species and sympatry of species. You have much more work to do before you could talk about species (morphological, bioacoustics, nuDNA analyses) and this should be acknowledged in the paper. The mtDNA data help raise testable hypotheses, but they don’t solve this complex of species. Furthermore discussion on the difficulties of mtDNA-based species hypotheses is missing. What about mtDNA ghost lineages? What about mtDNA introgression? This paper is quite light on references and this can be explained by lack of discussion of such evolutionary processes. Finally I think the biogeographical aspect of this paper should be fleshed out. Obviously in the set-up of the paper you want to explain why there is such a richness of (cryptic) species in the region and furthermore the data provide insights on biogeography. The authors seem to not have realized that the Istmus of Kra is an important biogeographical boundary and there is much digression in the discussion that can be ditched. These frogs once again validate this boundary and that is something to discuss in a wider context. I think the presentation of the mtDNA data can be better. The paper is quite well written but I do note some obvious issues. I give more details below.

The set-up of this work in the introduction is not optimal. This concerns the first and lines 81 – 83 from the last paragraph. These need to be combined as currently the flow of the introduction is cut up. I suggest first para the greater context of this study, second para the system (so as it is now) and third para the aims of the study (so lines 83-87). Regarding the set-up, I think it makes sense to start about SE Asia as a hotspot for amphibians. However, lines 50-53 are just a list of taxa with references and the link with the present work is not clear. This could be fleshed out however. I feel that this work is mostly about cryptic species and that is a good angle. It also makes sense to give some more background on the historical biogeography of the region to explain why there are so many species and to use new phylogenies such as created here to inform on historical biogeography.

About the study system: it is very difficult for me as a reader without knowledge on this system to grasp this. Are all the species you listed members of this complex and are all other known species of Polypedates not members of this complex. If so how was this determined? How are the relevant species distributed (perhaps provide a map with ranges and type localities?).

Line 45: remove substantial as hotspot already suggests biodiversity is substantial

Line 79: a complex is not a cryptic species, but a complex may contain cryptic species.

Line 49: remove “in the current period” or change to “observed today”

Line 54: Change “The clarification of ambiguous species” to “Accurate species delimitation”

Line 91: It is mentioned in the introduction that five (described) species of the complex are found in Thailand. But why are not all species included in this study? And which species are included? It is not clear to me how you chose the sampling that you did. Is it perhaps the case that some described species cannot be distinguished based on mtDNA?

Line 129: It seems like you directly provide a result here. In the Methods you should say you used Barrier 2.2 and why, but not what the outcome was, that should be in the Results.

Line 130-131: THE minimum spanning network >>> A minimum spanning network

Line 143: to assume should be assuming?

Line 144: Multiple should be multimodal?

Lines 161-164: It is not clear how you go from clades to species names. Do you base that on included sequences that were ascribed to a species? You need to be explicit. Of course you run the risk now of equating mtDNA with species while that is a dangerous interpretation without other data such as morphology and nuDNA. So you need to bring these names in carefully. It needs to be explained where the use of Polypedates sp. comes from. Does that mean this clade cannot be ascribed to a particular species while the others can? And if so, is that because you simply did not include relevant species or is it likely to be something new? It is not clear. I think it is not a good idea to talk in the Results about species rather than clades, for example describing the distribution of species based on mtDNA clades. If you want to interpret clades as species you would do that in the Discussion, but of course you need to be very, very careful there because mtDNA alone is not a good basis for species delineation.

Line 161: “Bayesian analysis model with MrModeltest” this phrasing makes no sense. You also did a Maximum Likelihood analyses. Don’t repeat methods.

Line 162: could consists >>> either just say consists or something that you recognize four clades.

Line 165: I guess you mean inference?

Line 174: incomplete sentence

Line 174-175: this is interpretation and does not belong in Results.

Line 176-177: This sentence is gibberish.

Line 181-186: there seems to be circularity here. You have allocated clades and hence haplotypes to species and now you say haplotypes belong to species. And what do you mean with unique haplotypes?

Line 202-203: Remove “one of the most notoriously challenging” as I can think of a million other difficult groups.

Line 204: You do not SOLVE anything, you get a little bit closer to the solution or at least to a hypothesis.

Line 205-208: These sentences should not be here. They just show that you should have used another marker than COI. But that is the marker you have so that is what you have to stick with. These studies may provide some insight into the system (I am not going to read them) and might be worth to integrate into the introduction of the study system, but there is no point of mentioning this here.

Lines 208-214: What you do here is just not right. You cannot just assume that these four clades are species and that these species occur sympatrically. You could only do that if you had independent data to delineate species, such as morphology or nuDNA. What about mtDNA introgression or simply the occurrence of distinct mtDNA clades in a single species? I am familiar with European newts and all these issues occur here. Related to this and hinted to in Line 221-222, apparently morphology could give some insights and you have looked at morphology but only as an aside? I noticed you have collected whole individuals so either you would do a morphological analyses of all these samples to see if you could come up with a morphological based species delineation to compare your mtDNA dataset to, or you do not make larger claims than that the mtDNA supports and can support.

Line 223-225 & 228-229: again, how do you link these names to these clades, it is not clear. Is it based on and ID given to a sequence taken from GenBank? If so you need to be clear on this (and what if the ID on GenBank is wrong?).

Lines 227-228: This is a strange sentence, it seems to suggest that somebody already looked at the exact same frogs and found the same pattern, but that is not the case. You mention another taxon where this pattern was found. Be more clear, so say it was stated in another taxon. However, there is much more to it than just a single other taxon, this is a known biogeographical boundary (also applies to lines 233-238). Isn’t this the official boundary of the Sunda region? This needs to be worked out because it is this kind of historical biogeographical inferences that are juicy results. Lines 243-247 seem to suggest that nobody before knew that this was a biogeographical barrier…

Lines 239-243: I don’t see what is the point of this section? So they could be moved by people (but I guess islands would also have been connected to the mainland during glacial times?) but how does that relate to the current work? You find a clear biogeographical boundary despite any artificial movement.

Lines 248-265: I don’t see the point of this section. There are two biogeographical regions and these frogs seem to show a divide as well, that is the bottom line (and for all the sympatric lineages you have no idea if they even are species and if so what separates them).

Lines 266-284: again you talk about a particular species without justification. Why would you need to invoke human-mediated translocation to explain homogeneity? Why not a recent population expansion from a small source population?

In the conclusion you talk about putative species and clades and this is the proper route to take.

Fig. 1 This legend is a mess.

It makes MUCH more sense to show clades instead of haplotypes in Fig. 2. This way the clades in Fig. 1 can be directly linked to the map and that is way more insightful. The present use of colours in Fig. 2 is poor, I cannot distinguish different shades of yellow etc. In Fig. 3 colouring the haplotype network according to populations is pretty pointless and actually very confusing. This haplotype network is better presented as part of Fig. 1. Just use the colours of clades for the haplotypes belonging to clades.

Reviewer 2 ·

Basic reporting

The grammar of the manuscript is not acceptable. There are many sections where I did not understand what the authors were trying to communicate. As an example, consider the first sentence of the abstract. Do that authors mean to imply that the taxonomic uncertainty in this group makes it challenging to infer its biogeographic history? I think so, but the wording is unclear and ambiguous in the first sentence. There are many additional examples.

The figures are clear and easy to interpret. The references are sufficient and appropriate.

Experimental design

The research is clearly within the scope of the journal.

The research question is not clearly define, due in large part to the ambiguities in the written presentation.

The methods description for the molecular biology are adequate.

The methodology for the data analysis is adequate in general. However, there are several small issues that should be addressed:
i. It is inconsistent to use a model of sequence evolution to infer the phylogeny (i.e., lines110-115), but to then calculate genetic distance for use in the other analyses using p-distances (line 128) because substitutions are not corrected in the p-distance as they are in the phylogeny inference. Thus, the same data can produce two different distance matrices.
ii. The Mantel test has been questioned on several levels for inferring isolation by distance. Better options are available (e.g., Legendre P, Fortin, MJ. 2010. Comparison of the Mantel test and alternative approaches for detecting complex multivariate relationships in the spatial analysis of genetic data. Mol. Ecol. Resour. 10, 831-844. AND Diniz-Filho JA, et al. 2013. Mantel test in population genetics. Genet. Mol. Biol. 36, 475-485.)
iii. It is in poor form to use a software package such as DNAsp to conduct an analysis such as the mismatch distribution without citing the original justification for this test [in this case Rogers, A. R., & Harpending, H. (1992). Population growth makes waves in the distribution of pairwise genetic differences. Molecular biology and evolution, 9(3), 552-569.]

Validity of the findings

The general findings appear valid.

The authors appear to have interpreted their results in a reasonable manner.

Additional comments

I believe that this work has a substantial amount of potential. However, I cannot recommend that it be published without a clear improvement in the presentation, and a correction of some of the analyses.

---

## Round 0.2 · Minor Revisions

The reviewer has made a number of very helpful suggestions, mainly to help the language and flow of the paper. For the information presented, I feel that the paper is rather long-winded, especially the Discussion. Try to simplify the language and remove extraneous speculation and detail.

Reviewer 1 ·

Basic reporting

No comment

Experimental design

No comment

Validity of the findings

No comment

Additional comments

Buddhachat and Suwannapoom revised their submission on a phylogeography of frogs belonging to the Polypedatus leucomystax complex occurring in Thailand. They have managed to improve the clarity of their MS but I still thought it somewhat long-winded in places. Below I suggest some cuts to streamline it more. I also have some language/style issues.

Why is the list of five species in line 54 not a subset of the six species mentioned in 50-51? Based on everything you wrote before that should be the case! Also, please keep the order of names consistent (e.g. alphabetical).

Line 51: delimited from >>> distinguished in

Line 56: remove redundant words “A study by”

Line 64: remove “highly”

Line 67-69: remove these sentences, they are out of place here.

Line 76: remove redundant words “In this study”

Line 90: remove “independently”

Line 91: genus name should be abbreviated, it seems you are inconsistent with this throughout the MS

Line 115: remove “population genetics software”

Line 142: could consist >>> consists

Line 149-150: the clade C population >>> clade C

Line 157-158: remove this sentence

Line 162-163: Change sentence to: Haplotypes F and G of clade D are considerably divergent from the rest of clade D and we partition it in two subclades: D1 and D2.

Line 184 and elsewhere: it is quite confusing to call this Northern B rather than Northern D…

Lines 190-198: These sentences are irrelevant, remove. I think spelling out a phylogeny rather than simply referring to the figure that summarizes the info is a bit much for Results already, but there is definitely no place for that in Discussion.

Line 198-199: Although mtDNA seemed to offer cleanly split clades with a unique haplotype in each clade, mtDNA introgession/hybridization between the closely related species can occur. This can
occur particularly in sympatric populations or at the boundaries of species distributions resulting
from incomplete reproductive isolation. >>> Although the mtDNA phylogeny reveals the presence of distinct clades, mtDNA introgression is frequently observed where closely related species are in secondary contact (CITE: (1) Toews DPL, Brelsford A. 2012. The biogeography of mitochondrial and nuclear discordance in animals. Mol. Ecol., 21:3907-3930. (2) Zieliński P, Nadachowska-Brzyska K, Wielstra B, Szkotak R, Covaciu-Marcov S, Cogălniceanu D, Babik W. 2013. No evidence for nuclear introgression despite complete mtDNA replacement in the Carpathian newt (Lissotriton montandoni). Mol. Ecol., 22:1884-1903. (3) Wielstra B, Burke T, Butlin RK, Arntzen JW. 2017. A signature of dynamic biogeography: enclaves indicate past species replacement. Proceedings of the Royal Society of London B: Biological Sciences, 284:20172014. (4) Wielstra B, Burke T, Butlin RK, Avcı A, Üzüm N, Bozkurt E, Olgun K, Arntzen JW. 2017. A genomic footprint of hybrid zone movement in crested newts. Evolution Letters, 1:93-101.)

Line 202-205: For instance, some populations of European newts Triturus montandoni have an mtDNA haplotype of Triturus vulgaris in an area where their distributions are connected, most likely as a result of historical or ongoing hybridization and multiple introgression of mtDNA from T. vulgaris to T. montandoni (Babik et al., 2005). – I think this sentence is too much detail and can be removed. The taxonomy used is outdated and a more recent and more detailed study is a more appropriate citation. See the rewritten sentence I propose above and I suggest to add some (newt) citations relevant citations there.

Line 187-189, 201-202 and 205-207: Merge these sentences and place them after suggested rewrite of lines 198-199 as “Therefore, the use of mitochondria DNA data alone is not sufficient for species delineation. An integrative approach, consulting additional data such as morphology, bioacoustics, ecology, and/or nuclear DNA, would be required to more accurately assess the taxonomy of the Polypedates leucomystax complex (CITE: Padial J, Miralles A, De la Riva I, Vences M. 2010. The integrative future of taxonomy. Front. Zool., 7:16.)”.

Line 207: remove “A recent study by”

Line 207: THE P. leucomystax complex

Line 217: remove “expansion”

Line 212-213: “forms a significant region separating them” >>> “corresponds to a considerable phylogeographic break”

Line 219-220: “we assumed that the northern B clade and the Southern clade might be P. megacephalus and P. leucomystax, respectively” we ascribe the northern B clade to P. megacephalus and the southern clade to P. leucomystax”

Line 221: remarkable >>> considerable

Line 223: to suggest a hypothesis >>> to suggest OR to hypothesize

Line 231: remove “A study by”

Line 236-238 “Additionally, data on rainfall in Thailand by the Thai Meteorological Department indicates a difference in the amount of rainfall between areas to the north and south of the isthmus (Fig. 3).” This does not add anything, remove this sentence and remove the Fig. 3 (which is not convincing).

Line 238: Based on this, we believe that climate may be a >>> We suggest climate was a

Line 240-241: remove this sentence

Line 245: Where does this molecular dating come from? I presume a previous study? Cite it!

Line 247: may be >>> is

Line 250-262: Remove this paragraph. You have linked the phylogeographic break at the IOK to correspond to distinct climate zones, that is enough. This paragraph is highly speculative and confusing and does not help.

Line 272: remove “great”

---

## Round 0.3 · Minor Revisions

One again, I would ask that you get an English speaker to read your entire manuscript to check the grammar. I see the following sentence, for example, that makes no linguistic sense:

"For instance, a pair of crested newt species (genus Triturus) in hybrid zone (the south of the Marmara sea) exists the introgression and support the hybrid zone movement, leading to insight into historical biogeography and the speciation process (Wielstra et al., 2017b)."

I suggest replacing with:
"For instance, a pair of crested newt species (genus Triturus) form a hybrid zone (south of the Marmara Sea), where introgression suggests movement of the hybrid zone (Wielstra et al., 2017b)."

---

## Round 0.4 · accepted · Accept

Thank you for attending to the problem sentence.